# Fosfomycin, Applying Known Methods and Remedies to A New Era

**DOI:** 10.3390/diseases8030031

**Published:** 2020-08-07

**Authors:** Viviana Albán M., Estefanía Mariño-Brito, Fernando Villavicencio, Carolina Satán, José E. Villacís, Mónica C. Gestal

**Affiliations:** 1Centro de Referencia Nacional de Resistencia a los Antimicrobianos, Instituto Nacional de Investigación en Salud Pública “Leopoldo Izquieta Pérez”, Quito 170403, Ecuador; albanmviviana@gmail.com (V.A.M.); fervillavicencioz@gmail.com (F.V.); caro.cess.2810@gmail.com (C.S.); 2Instituto de Microbiología, Universidad San Francisco de Quito, Quito 170901, Ecuador; 3Facultad de Medicina, Pontificia Universidad Católica del Ecuador, Quito 170143, Ecuador; kemarino93@gmail.com; 4Health Science Center, Louisiana State University (LSU), Shreveport, LA 71103, USA

**Keywords:** carbapenem-resistant *Enterobacteriaceae* (CRE), fosfomycin, agar dilution, disk diffusion

## Abstract

The exponential increase in the numbers of isolates of Carbapenem-Resistant *Enterobacteriaceae* (CRE) creates the need for using novel therapeutic approaches to save the lives of patients. Fosfomycin has long been considered a rational option for the treatment of CRE to be used as part of a combined therapy scheme. However, the assessment of fosfomycin susceptibility in the laboratory presents a great challenge due to the discrepancies found between different methodologies. Thus, our goal was to evaluate fosfomycin susceptibility in a group of 150 *Enterobacteriaceae* bacterial isolates using agar dilution as the gold standard technique to compare the results with those obtained by disk diffusion. We found a fosfomycin susceptibility of 79.3% in general terms. By comparing both methodologies, we reported a categorical agreement of 96% without Very Major Errors (VMEs) or Major Errors (MEs) and 4% of minor Errors (mEs). Our results suggest that fosfomycin could provide a rational alternative treatment for those patients that are infected by a Multidrug-Resistant (MDR) microorganism that is currently untreatable and that the disk diffusion and classical agar dilution techniques are adequate to assess the resistance profile of CRE to fosfomycin.

## 1. Introduction

The increasing number of isolates of Carbapenem-Resistant *Enterobacteriaceae* (CRE) in diagnostic laboratories is one of the greatest concerns in public health because of its high-risk association with morbidity and mortality [1]. Multidrug-Resistant (MDR) bacteria are defined as bacteria that show no susceptibility to at least one antibiotic type from more than three different antimicrobial families. The exponential increase of MDR isolates worldwide creates an imperative need for novel therapeutic strategies [2]. There are a number of different approaches which are being investigated, including immunotherapies, the use of proteomics or nanotechnology, and the rediscovery of old antimicrobials such as the revival of fosfomycin [3].

Fosfomycin irreversibly blocks the condensation of UDP-N-acetyl glucosamine with p-enolpyruvate by covalently binding to the enzyme UDP-GlcNAc enolpyruvyl transferase (MurA), causing the inhibition of peptidoglycan biosynthesis and consequent bacterial cell death [4]. Moreover, fosfomycin is considered a good elective treatment against a broad spectrum of Gram-negative and Gram-positive bacteria, including CRE, providing a great alternative for its use in the treatment of various infectious diseases [5]. Amongst the great properties of this antibiotic for its use in clinical practice are: its excellent distribution in almost every tissue [5], its undetectable levels of cytotoxicity, and its capacity for being easily secreted from the body [6,7]. Although, in general, this drug has been considered safe, after intravenous administration, hypokalemia and sodium overload have been reported [8].

Unfortunately, currently, the use of fosfomycin is mostly indicated for uncomplicated urinary tract infections [9], overlooking its great immunomodulatory properties. Fosfomycin can act to promote phagocyte-mediated killing [10], block ERK/P38-mediated NLRP3 inflammasome activation by a-hemolysin [11], promote bactericidal activity in neutrophils [12] via the kinase c-signaling pathway [13], and has immunomodulatory activity in adaptive responses [14,15]. Excitingly, fosfomycin has demonstrated that it can modulate metabolism, affecting several characteristic bacterial phenotypes [16], as well as hosts [17,18]. Despite all the good qualities of this antibiotic, its use in combined therapy for the management of MDR Gram-negative bacterial infections [19], including respiratory, subcutaneous, muscular, bone, gastrointestinal, nervous system, and systemic infections [7,20], has been sporadic.

One of the main constraints on the use of fosfomycin in clinical practice is given by the lack of a cheap, easy, and rapid method to evaluate the resistance of bacterial isolates against this antibiotic. The Clinical Laboratory Standard Institute (CLSI) approved agar dilution supplemented with 25 mg/L of glucose-6-phosphate (G6P) as the gold standard technique for testing fosfomycin in *Escherichia coli* isolates recovered from urinary tract infections. Additionally, the CLSI has established breakpoints for disk diffusion using 200 μg fosfomycin plus 50 μg of G6P disks [9]. Although the CLSI has determined breakpoints for Minimal Inhibitory Concentration (MIC) and Zone Diameter for urinary isolates of *E. coli*, the European Committee on Antimicrobial Susceptibility Testing (EUCAST) has recommended lower breakpoint values following agar dilution and disk diffusion supplemented with G6P for *Enterobacteriaceae* and *Pseudomonas aeruginosa* [21]. It is noteworthy to highlight that discrepancies between agar dilution and broth dilution MICs for fosfomycin have been reported [22], creating a conundrum for the use of microdilution as the standard method in clinical diagnostic labs.

Thus, the aim of this study was to assess the in vitro activity of disodium fosfomycin in CRE strains isolated in Ecuador using agar dilution technique as the gold standard and compare this with disk diffusion, with the goal of generating some guidelines for a fosfomycin susceptibility test that can guide clinicians in their therapeutic choices. We expect that this simple manuscript will convince laboratories to test old antibiotics using current gold standards and old-fashioned tests. Our aim is to also push researchers to take advantage of proteomics data availiable and use them to develop novel strategies to test susceptibility profiles in laboratories with reduced budgets, which generally match those that suffer more from antibiotic resistance.

## 2. Materials and Methods

### 2.1. Bacterial Isolates

A total of 150 CRE collected from clinical isolates between 2017 and 2018 were randomly selected from the National Reference Center for Antimicrobial Resistance (CRN-RAM) strain collection. The bacterial isolates included in this study were *Klebsiella pneumoniae* (125), *Citrobacter freundii* (6), *Escherichia coli* (4), *Klebsiella oxytoca* (4), *Enterobacter cloacae* (3), *Klebsiella aerogenes* (2), *Morganella morganii* (2), *Serratia marcescens* (2), *Citrobacter youngae* (1), and *Klebsiella ozaenae* (1). The identification of all the strains was confirmed using VITEK 2 GN cards. An inclusion criterion was that all strains were resistant to carbapenems.

### 2.2. Susceptibility Testing

Antimicrobial susceptibility profiles were obtained via VITEK 2 using N272 cards. Fosfomycin susceptibility was determined in duplicate using Disk Diffusion (DD) and Agar Dilution (AD) methods. DD was performed on Mueller–Hinton agar [23] (Difco^TM^) using 200 μg fosfomycin/50 μg G-6-P disks (Oxoid). AD was done on Mueller–Hinton agar with fosfomycin disodium salt (Sigma Chemical Co., St Louis, MO, USA) supplemented with 25 mg/L of glucose-6-phosphate (G6P) (Sigma Chemical Co., St Louis, MO, USA). Both techniques were executed according to the CLSI guidelines M02-A12 and M07-A10, respectively. A two-fold dilution method from 16 to 256 μg/mL was used to evaluate MICs results. Susceptibility results were interpreted according to the CLSI M100 2018 breakpoints for *E. coli* extrapolated to *Enterobacteriaceae* (DD: ≥16 susceptible, 13-15 intermediate, and ≤12 resistant; AD: ≤64 susceptible, 128 intermediate, and ≥256 resistant). *Escherichia coli* ATCC^®^ 25922 was used as a quality control strain.

### 2.3. Molecular Detection of Carbapenem Resistance

All strains were screened by end point Polymerase Chain Reaction (PCR) to identify the presence of genes that confer resistance to carbapenem, including *blaOXA-48*, *blaVIM*, *blaKPC*, *blaNDM*, and *blaIMP*. The amplification conditions have been described elsewhere [24].

### 2.4. Statistical Analysis

The strength of agreement analysis between both methodologies was establish through Cohen’s kappa index using RStudio version 1.3.959 [25]. Additionally, we used a categorical agreement, which is a concept that depends on the breakpoint interpretation (susceptible, intermediate, or resistant) categorical match. Furthermore, the existing discrepancies between methodologies were categorized as Very Major Errors (VMEs) when disk diffusion revealed susceptibility while the gold standard concluded resistant, Major Errors (MEs) when disk diffusion revealed resistant while the gold standard concluded susceptible, and minor Errors (mEs) when disk diffusion revealed intermediate susceptibility while the gold standard concluded susceptible or resistant.

## 3. Results

### 3.1. Fosfomycin Revealed Great In Vitro Efficacy Against Carbapenem-Resistant Enterobacteriaceae

For all CRE isolates selected for this study, 98.67% harbored a *blaKPC* gene and 1.33% a *blaNDM* gene. Interestingly, we observed resistance to more than three independent antibiotic categories in all isolates and allocated them into the classification of MDR bacterial strains. The fact that the resistance was against different antibiotic families indicated that these strains harbor multiple mechanisms of resistance. Interestingly, we found a high level of colistin resistance amongst *Klebsiella pneumoniae* isolates (Appendix A). This presents a new problem, as colistin would not be an appropriate therapeutic alternative for patients. To our surprise, when assessing fosfomycin susceptibility by agar dilution (Table 1), the results revealed 79.3% of susceptibility, 2.7% of intermediate resistance, and 18.0% of resistance amongst these isolates, suggesting that fosfomycin could provide a novel substitute for the treatment of patients.

### 3.2. Disk Diffusion is A Reliable Test to Evaluate Fosfomycin Resistance Similar to the Agar Dilution Technique

To evaluate the harmony between both methods, we applied the Cohen’s kappa index to correlate the MIC values and inhibition zones interpreted by CLSI breakpoints. Our results revealed an exceptional agreement between both techniques (κ = 0.88) following the Landis and Koch classification scale (Table 2). Additionally, we obtained a categorical agreement of 96%, leaving 4% inconsistencies between methodologies, which were categorized as minor Errors (mEs). It is important to note that the discrepancies we identified corresponded when one of the methods revealed intermediate susceptibility while the other concluded susceptible or resistant. No false sensitive (VME) or false resistant (ME) error results were found (Figure 1). Overall, our results demonstrate that disk diffusion can be used as a reliable method to quickly assess fosfomycin resistance in clinical diagnostic laboratories.

## 4. Discussion

According to the World Health Organization Antimicrobial Resistance Global Report on Surveillance in 2014, *Klebsiella pneumoniae*, a CRE in the region of the Americas, appears in a total of 56% of all bacterial isolates of *Klebsiella* species [26]. In Ecuador, *Enterobacteriaceae* carbapenem resistance is of great concern, representing 16.5% of bacterial isolates from hospitalized patients (data not published). Colistin has been considered one of the last alternatives for the treatment of CRE infections [27], however, high rates of resistance development have been reported worldwide and the recent discovery of a plasmidic mechanism of resistance that can easily spread (*mcr*) increases the concern [28]. New molecules, such as ceftazidime/avibactam (CAZ/AVI), have appeared and their use is associated with an outstanding in vitro activity against *Enterobacteriaceae* carbapenemase producers isolates, especially *bla-KPC* and *bla-OXA-48*, as well as a significant decrease in hospital-associated mortality 30 days after starting treatment [29]. Nevertheless, although CAZ/AVI has exhibited excellent outcomes in terms of quality-adjusted life-years, costs, and incremental cost-effectiveness ratios [30], some countries like Ecuador have not yet included CAZ/AVI in the national drug list to be used in clinical settings. Thus, the rediscovery of old antibiotics, such as fosfomycin, is considered an important antibiotic treatment strategy.

Despite the low usage of fosfomycin in clinical settings, this antibiotic has been used in a combined therapy with carbapenems, tigecycline, and aminoglycosides [19] in some trials, rendering rates of clearance of >80% [31]. The concern remains to be able to test the resistance profile in vitro without highly sophisticated equipment that can be only be acquired in high-end laboratories. Susceptibility tests play a key role in guiding clinical personnel when deciding the best therapy in each case, and the lack of validated, cheap, and easy methods to evaluate fosfomycin resistance remains a challenge for its use in clinical settings. In this manuscript, we evaluate the susceptibility profile against fosfomycin of a group of 150 CRE isolates and our results indicate that nearly 80% of the isolates remain susceptible to fosfomycin in vitro. A previously published study suggested that fosfomycin resistance increased 11% in Ecuador between 2002 and 2012 [32]. Previous investigations reported great variation in the susceptibility profile, ranging from 39% to 100% when applying the agar dilution technique [33]. Overall, these results suggest that more clear indications on the testing methods, as well as interpretation, need to be made.

To evaluate the use of an alternative technique in clinical settings, including the results obtained by disk diffusion and agar dilution, the CLSI recommends <10% mE, <3% ME, and 1.5% VM scores in order to validate the performance of a specific technique to be used in susceptibility tests [34]. Our findings rendered an excellent agreement between the agar dilution and disk diffusion techniques, with a Kappa index of 0.88, taking into account the MIC values and inhibition zones in terms of categorical interpretation. Susceptible, intermediate, or resistant are the three analyzed categories according to the CLSI M100 guideline. Additionally, we report a categorical agreement of 96% with no VMEs or MEs found and 4% of mEs that could be due to the presence of colonies within the inhibition zone that could complicate the measurement. These findings are consistent with a previous study of *Klebsiella pneumoniae bla-KPC*, including tigecycline and colistin resistant isolates, in which the authors reported a categorical agreement of 66.6% without VMEs or MEs but with a 33.8% rate of mEs explained by the numerous isolates that showed inhibitory diameters of 15, 14, and 13 mm but were susceptible by agar dilution [35]. Additionally, the investigation led by Lu et al. also proposed that the disk diffusion technique could be considered an alternative method to assess fosfomycin susceptibility based on the low rate of MEs (4%) and mEs (9%) found in their study [36].

Kasse et al. [37] and Van den Bijllaardt et al. [38] reported VME rates that varied between 16.7% and 12.9%, respectively, in *Enterobacteriaceae* isolates when comparing disk diffusion and agar dilution for fosfomycin using the breakpoints suggested by Pasteran et al. (susceptible: ≥17) [39] and Epidemiological cut-off values (ECOFFs) of 64 mg/L, concluding that disk diffusion could be overestimating the real MIC value. In order to solve these discrepancies, Mojica et al. suggests the interpretation of disk diffusion results in accordance with CLSI established breakpoint taking into account EUCAST recommendations for obtaining the inhibition zone diameter to reach the best performance of DD [22].

To summarize, by an in vitro test, we found that fosfomycin could provide a rational alternative treatment for those patients that are infected by an MDR microorganism that is currently untreatable. This old antibiotic is gaining attention for its efficacy, as well as for its immunomodulatory properties, and the greatest limitation, the availability of laboratory tests to assess the susceptibility of the different bacterial isolates. Our results indicate that fosfomycin might be efficient even against those colistin-resistant isolates for which we do not have any alternative treatment. This has great implications, as for many patients, there is no other feasible therapeutic option, and this can mean a chance for them. By comparing the two methodologies, our findings demonstrate that the disk diffusion and classical agar dilution techniques are adequate to assess the resistance profile of *Enterobacteriaceae* to fosfomycin. Nevertheless, it is important to also note our results suggest that isolates with intermediate results must be sent to a reference laboratory to be confirmed. Although, this might change when more isolates are studied or more work from other groups gets published. Additionally, most of our isolates were *Klebsiella pneumoniae* and, in other microorganisms, the agreement between both techniques may need to be reassessed. Overall, we believe that is important to use our results as guidance for future work and to promote the use of fosfomycin and disk diffusion tests in patients colonized/infected with MDR bacteria.

## Figures and Tables

**Figure 1 diseases-08-00031-f001:**
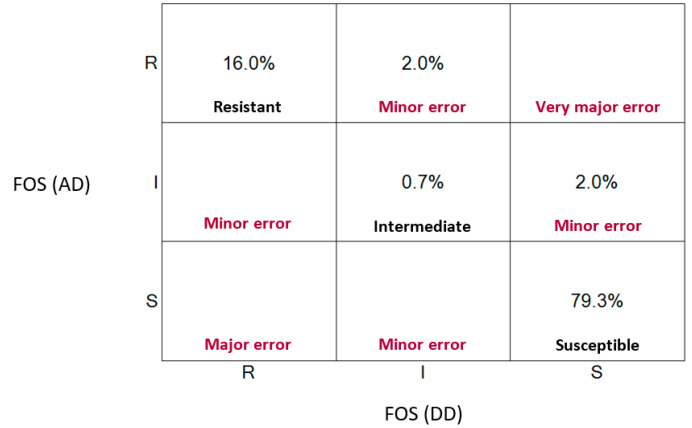
Scatter plot representing clinical errors between disk diffusion and agar technique.

**Table 1 diseases-08-00031-t001:** Fosfomycin Minimum Inhibitory Concentration (MIC) distribution in all isolates.

		Fosfomycin Agar Dilution				
Strains	n	Susceptible (S)			Intermediate (I)	Resistant (R)
		16 µg/mL	32 µg/mL	64 µg/mL	128 µg/mL	256 µg/mL
*Klebsiella pneumoniae*	125	46	37	15	3	24
*Klebsiella oxytoca*	4	4				
*Klebsiella ozaenae*	1			1		
*Escherichia coli*	4	4				
*Serratia marcescens*	2		2			
*Morganella morganii*	2				1	1
*Citrobacter freundii*	6	5				1
*Enterobacter cloacae*	3	2				1
*Enterobacter aerogenes*	2	1	1			
*Citrobacter youngae*	1	1				
TOTAL	150	63	40	16	4	27

**Table 2 diseases-08-00031-t002:** Agreement between agar dilution and disk diffusion methods.

Species	n	Agar Dilution			Disk Diffusion		
		S	I	R	S	I	R
*Klebsiella pneumoniae*	125	98	3	24	100	4	21
*Klebsiella oxytoca*	4	4	0	0	4	0	0
*Klebsiella ozaenae*	1	1	0	0	1	0	0
*Escherichia coli*	4	4	0	0	4	0	0
*Serratia marcescens*	2	2	0	0	2	0	0
*Morganella morganii*	2	0	1	1	1	0	1
*Citrobacter freundii*	6	5	0	1	5	0	1
*Enterobacter cloacae*	3	2	0	1	2	0	1
*Enterobacter aerogenes*	2	2	0	0	2	0	0
*Citrobacter youngae*	1	1	0	0	1	0	0
Cohen’s kappa index: 0.88 *							

* Strength of agreement based on Landis and Koch. Poor κ < 20, fair κ = 0.21–0.40, moderate κ = 0.41–0.60, good κ = 0.61–0.80, and very good κ = 0.81–1.00.

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
