# Peer review of "Fosfomycin, Applying Known Methods and Remedies to A New Era"

_diseases, 2020, doi:10.3390/diseases8030031_

Round 1
Reviewer 1 Report
For referencing purposes for other labs (and interest), it would be very useful to include a succinct summary of additional typing/isolate information, e.g. serotype, etc, for the 150 Enterobacteriaceae CRE collected from clinical isolates in a Table, and to include this information/Table within Supplementary materials.
In the Introduction (lines 64-76) the known adverse effects/events of fosfomycin could be very briefly mentioned, e.g. those discussed from Falagas et al., (2016) PMID: 26960938.
Author Response
Reviewer 1
We would like to thank the reviewer for the effort and detailed comments. This suggestions have significantly improved the manuscript.
Comment 1: For referencing purposes for other labs (and interest), it would be very useful to include a succinct summary of additional typing/isolate information, e.g. serotype, etc, for the 150 Enterobacteriaceae CRE collected from clinical isolates in a Table, and to include this information/Table within Supplementary materials.
Response 1: We would like to thank the reviewer for the comment. Unfortunately, these strains were collected between 2017 and 2018 and we have no access to them any longer. We decided not to do the serotyping at the time of the study, because we considered it was not within the scope of the manuscript; and although we regret it now, we cannot longer do it. However, we fully agree with the reviewer that this will be very important information and if we do a similar project in the future, we will definitely include that experiment in to our to do list.
Comment 2: In the Introduction (lines 64-76) the known adverse effects/events of fosfomycin could be very briefly mentioned, e.g. those discussed from Falagas et al., (2016) PMID: 26960938.
Response 2: We have included it in Line 52- “Although, in general this drug has been considered safe, after intravenous administration, hypokalemia and sodium overload have been reported”
Reviewer 2 Report
The manuscript addresses an important issue of the possibility of using fosfomycin for the treatment of infections caused by carbapenem-resistant Enterobacteriaceae. In recent years, there have been several publications on the sensitivity of these bacteria to fosfomycin, including the publication by Kaase et al. (2014) based on the same methodology as used by the authors of this manuscript. The publication is well written but has some shortcomings. First, the authors did not use a reference strain as a quality control (e.g., CLSI recommended Escherichia coli ATCC 25922) in the antimicrobial susceptibility test. There are some errors in the text, mainly editorial, which require correction:
the name "Enterobacteriaceae" and "in vitro" should be written in italics - throughout the text
L17 - no need to write "Carbapenem Resistant" (L17, 47) and "Fosfomycin" (L64) in capital letters
L26 - the use of unexplained abbreviations (VME, ME) in the Abstract is not recommended
CRE abbreviation is explained twice in the text (L17 and L47)
L67, 69, 205- no space before mg and ug
L89- remove "Enterobacteriaceae", CRE enough
remove spaces after ug (L106) and before 256 (L109)
L120-124 - the methodology for determining VME, ME and mE should be better described
remove the dot before [29] (L169) and before [22] (L209)
L202 & 206- correct the citation (remove M.) and put a dot after et al.
Author Response
Reviewer 2
We would like to thank the reviewer for all the attention put into the revision. Thanks for all the details that have made this manuscript a better paper. We provide the detailed answers below, and again thanks for all the suggestions.
Comment 1: The authors did not use a reference strain as a quality control (e.g., CLSI recommended Escherichia coli ATCC 25922) in the antimicrobial susceptibility test
Response 1: L109- We have included the quality control strain used: “Escherichia coli ATCC® 25922 was used as a quality control strain.”
Comment 2: The name "Enterobacteriaceae" and "in vitro" should be written in italics - throughout the text
Response 2: This has been revised as requested
Comment 3: L17 - no need to write "Carbapenem Resistant" (L17, 47) and "Fosfomycin" (L64) in capital letters
Response 3: Revised as requested. We consider that capital letters for Carbapenem Resistant Enteroacteriaceae in L17 are needed because is the first time we mention the term CRE and we believe this will avoid any further confusion.
Comment 4: L26 - the use of unexplained abbreviations (VME, ME) in the Abstract is not recommended
Response 4: L26- Revised as requested: “By comparing both methodologies, we found a categorical agreement of 96% without Very Major Errors (VME) or Major Errors (ME) and 4% of minor Errors (mE) recorded”
Comment 5: CRE abbreviation is explained twice in the text (L17 and L47)
Response 5: L-47- Revised as requested. “Moreover, fosfomycin is considered a good elective treatment against a broad spectrum of Gram-negative and Gram-positive bacteria including CRE”
Comment 6: L67, 69, 205- no space before mg and ug
Response 6: Revised as requested
Comment 7: L89- remove "Enterobacteriaceae", CRE enough
Response 7: Revised as requested
Comment 8: Remove spaces after ug (L106) and before 256 (L109)
Response 8: Revised as requested
Comment 9: L120-124 - the methodology for determining VME, ME and mE should be better described
Response 9: We included a more specific description for these terms in L-25:
“Furthermore, the existing discrepancies between methodologies were categorized as very major errors (VME) when disk diffusion revealed a sensitive result while the gold standard concluded resistant, major errors (ME) when disk diffusion revealed a resistant result while the gold standard concluded susceptible and minor error (mE) when disk diffusion revealed intermediate susceptibility while the gold standard concluded susceptible or resistant.”
Comment 10: Remove the dot before [29] (L169) and before [22] (L209)
Response 10: Revised as requested
Comment 11: L202 & 206- correct the citation (remove M.) and put a dot after et al.
Response 11: Revised as requested